# Elastic Mixture of Rank-Wise Experts for Knowledge Reuse in Federated Fine-Tuning

## Abstract

Federated fine-tuning offers a promising solution for adapting Large Language Models (LLMs) to downstream tasks while safeguarding data privacy. However, its high computational and communication demands hinder its deployment on resource-constrained devices. In this paper, we propose SMARTFED, a resource-efficient federated fine-tuning framework. SMARTFED intelligently reuses knowledge embedded in existing LoRA modules, eliminating the need for expensive training from scratch when adapting LLMs to new tasks. To effectively exploit this knowledge and ensure scalability, we introduce the Mixture of Rank-Wise Experts (MoRE). MoRE decomposes LoRA modules into fine-grained rank-level experts. These experts are selectively activated and combined based on input semantics and resource budgets. Moreover, to optimize resource utilization, we present the Elastic Expert Quota Allocation (EEQA). EEQA adaptively allocates expert capacity across parameter matrices based on their contribution to model performance, focusing computing resources on the critical experts. Extensive evaluations across multiple benchmarks demonstrate that SMARTFED significantly outperforms existing methods in model performance and training efficiency.

## 1 Introduction

Large Language Models (LLMs) (Guo et al., 2025; Bai et al., 2023) have demonstrated impressive performance across diverse tasks, with fine-tuning enabling alignment to task-specific objectives (Anisuzzaman et al., 2024). However, downstream data is often distributed across devices and subject to strict privacy regulations (e.g., GDPR), making centralized fine-tuning infeasible (Wu et al., 2025). Federated fine-tuning (Zhang et al., 2024) offers a compelling alternative by enabling collaborative model adaptation while preserving data privacy. While promising, the sheer scale of LLMs renders full-parameter fine-tuning prohibitively expensive for resource-constrained devices.

To bridge this gap, various parameter-efficient federated fine-tuning methods have been proposed (Bian et al., 2025), with LoRA (Hu et al., 2022) demonstrating superior efficiency and performance. However, existing methods typically train LoRA from scratch (Babakniya et al., 2023) (Figure 1(a)), requiring hundreds of communication rounds to converge and incurring substantial resource costs. With the growing availability of LoRA modules fine-tuned on diverse tasks (Huang et al., 2023), a natural question arises: *Can we just reuse existing LoRA modules to adapt LLMs to new tasks, thereby avoiding costly LoRA retraining and minimizing resource overhead?*

In this paper, we propose SMARTFED, a resource-efficient federated fine-tuning framework that intelligently reuses the knowledge embedded in existing LoRA modules to adapt LLMs to new tasks. The server first retrieves task-relevant modules from public repositories (e.g., LoRAHub (Huang et al., 2023)), and a trainable router dynamically activates them based on input semantics to enable context-aware knowledge composition for task-specific adaptation (Figure 1(b)). Thus, edge devices optimize only the router, avoiding costly LoRA retraining. However, directly fusing entire LoRA outputs is coarse-grained, and as more modules are retrieved, computational overhead also increases.

In response, we introduce the Mixture of Rank-Wise Experts (MoRE), which decomposes each LoRA module along the rank dimension, treating the paired vectors from the $\mathbf{A}$ and $\mathbf{B}$ matrices as individual experts. This architectural design enables fine-grained knowledge fusion and flexible rank activation to meet computational constraints. However, since different parameter matrices contribute unevenly to model performance, uniform expert quota allocation is suboptimal. We therefore

(a) Classic Federated Fine-Tuning Framework.

(b) Our SmartFed Framework.

Figure 1: Illustration of the classic federated fine-tuning framework versus our SMARTFED. Classic methods train LoRA modules from scratch (a), while SMARTFED updates only the router (b).

propose Elastic Expert Quota Allocation (EEQA), which adaptively assigns the number of activated experts for each parameter matrix according to its contribution, thereby prioritizing computing resources on key experts to maximize knowledge utility without increasing computational cost.

To validate the effectiveness of SMARTFED, we conduct extensive experiments on three representative LLMs, covering benchmarks across diverse domains. The results show that SMARTFED achieves up to 10.21% average performance gains, 3.95× faster convergence, 31.47× lower communication overhead, and 3.61× reduced energy consumption compared to existing methods. Further analysis highlights that SMARTFED demonstrates superior data efficiency; for instance, on math-word problems, it surpasses baselines by 8.02% while utilizing only 10% of the training data versus 100% for baselines. This characteristic is particularly valuable for data-scarce domains.

## 2 BACKGROUND AND MOTIVATION

### 2.1 LoRA BASICS

LoRA (Hu et al., 2022) is a parameter-efficient fine-tuning method that injects trainable low-rank matrices into pre-trained weights, enabling task adaptation without modifying original parameters. Specifically, for the weight matrix $\mathbf{W}_0 \in \mathbb{R}^{d \times d}$, LoRA introduces a low-rank update $\Delta \mathbf{W} = \mathbf{B}\mathbf{A}$, where $\mathbf{A} \in \mathbb{R}^{r \times d}$ and $\mathbf{B} \in \mathbb{R}^{d \times r}$ with $r \ll d$. For an input $\mathbf{x} \in \mathbb{R}^d$, the forward computation is:

$$\mathbf{h}' = \mathbf{W}_0 \mathbf{x} + \Delta \mathbf{W} \mathbf{x} = \mathbf{W}_0 \mathbf{x} + \mathbf{B}\mathbf{A}\mathbf{x}, \tag{1}$$

where $\mathbf{h}'$ denotes the adapted output, and $\mathbf{B}\mathbf{A}\mathbf{x}$ encodes task-specific information. By introducing only a small number of trainable parameters, LoRA substantially reduces fine-tuning costs. With the growing availability of open-source LoRA modules fine-tuned on diverse tasks, reusing their embedded knowledge without retraining presents a promising avenue for further cost reduction.

### 2.2 LIMITATIONS OF EXISTING LoRA REUSE METHODS

A straightforward approach for reusing task-adapted LoRA modules is to combine them via linear arithmetic (Ilharco et al., 2022), synthesizing a new module that integrates knowledge from multiple tasks. For example, Zhang et al. (2023a) achieve this by performing element-wise addition over the corresponding parameter matrices (Figure 2(a)). Formally, given a set of LoRA modules $\{(\mathbf{B}_1, \mathbf{A}_1), \ldots, (\mathbf{B}_N, \mathbf{A}_N)\}$ fine-tuned on $N$ distinct tasks, the composition is defined as:

$$\mathbf{B}_{\text{add}} = \sum_{n=1}^{N} \lambda_n \mathbf{B}_n, \quad \mathbf{A}_{\text{add}} = \sum_{n=1}^{N} \lambda_n \mathbf{A}_n, \tag{2}$$

where $\lambda_n$ is a weighting coefficient controlling each module's contribution. While simple and computationally efficient, this approach requires all LoRA modules to share identical dimensions and may introduce substantial noise, particularly when the constituent modules are trained on semantically diverse or conflicting tasks. For instance, consider two LoRA modules $(\mathbf{B}_1, \mathbf{A}_1)$ and $(\mathbf{B}_2, \mathbf{A}_2)$, which generate the following residual updates:

$$\Delta \mathbf{h}_1 = \mathbf{B}_1 \mathbf{A}_1 \mathbf{x}, \quad \Delta \mathbf{h}_2 = \mathbf{B}_2 \mathbf{A}_2 \mathbf{x}. \tag{3}$$

Ideally, to integrate the knowledge from both tasks, the final output should be:

$$\mathbf{h}' = \mathbf{W}_0 \mathbf{x} + \lambda_1 \Delta \mathbf{h}_1 + \lambda_2 \Delta \mathbf{h}_2 = \mathbf{W}_0 \mathbf{x} + \lambda_1 \mathbf{B}_1 \mathbf{A}_1 \mathbf{x} + \lambda_2 \mathbf{B}_2 \mathbf{A}_2 \mathbf{x}. \tag{4}$$

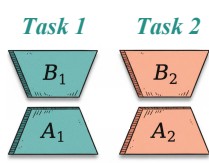
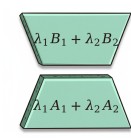
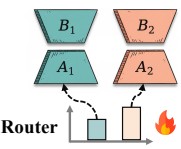
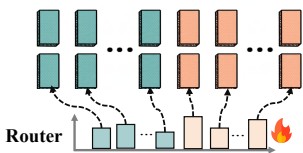

Task-Specific LoRA    (a) Linear Arithmetic.    (b) Mixture of LoRA Experts.    (c) Mixture of Rank-Wise Experts.

Figure 2: Comparison of three knowledge reuse strategies using two task-specific LoRA modules: (a) synthesizing a single LoRA module through linear arithmetic, (b) integrating information from entire LoRA modules, and (c) integrating information from rank-wise components.

Under naive parameter merging, as defined in Equation 2, the LoRA modules are combined as:

$$\mathbf{B}_{\text{add}} = \lambda_1 \mathbf{B}_1 + \lambda_2 \mathbf{B}_2, \quad \mathbf{A}_{\text{add}} = \lambda_1 \mathbf{A}_1 + \lambda_2 \mathbf{A}_2, \tag{5}$$

which yields the following output:

$$
\begin{aligned}
\mathbf{h}' &= \mathbf{W}_0\mathbf{x} + \mathbf{B}_{\text{add}}\mathbf{A}_{\text{add}}\mathbf{x} \\
&= \mathbf{W}_0\mathbf{x} + (\lambda_1\mathbf{B}_1 + \lambda_2\mathbf{B}_2)(\lambda_1\mathbf{A}_1 + \lambda_2\mathbf{A}_2)\mathbf{x} \\
&= \mathbf{W}_0\mathbf{x} + \underbrace{\lambda_1^2\mathbf{B}_1\mathbf{A}_1\mathbf{x} + \lambda_2^2\mathbf{B}_2\mathbf{A}_2\mathbf{x}}_{\text{task-specific adaptation}} + \underbrace{\lambda_1\lambda_2\mathbf{B}_1\mathbf{A}_2\mathbf{x} + \lambda_1\lambda_2\mathbf{B}_2\mathbf{A}_1\mathbf{x}}_{\text{cross-task interference (noise terms)}}.
\end{aligned}
$$

These cross-task terms, $\lambda_1\lambda_2\mathbf{B}_1\mathbf{A}_2\mathbf{x}$ and $\lambda_1\lambda_2\mathbf{B}_2\mathbf{A}_1\mathbf{x}$, deviate from task-specific directions and can cause destructive interference from parameter conflicts, leading to information loss. Consequently, this approach struggles to integrate knowledge across multiple LoRA modules. An alternative, Mixture of LoRA Experts (MoLE) (Wu et al., 2024), treats each task-specific LoRA module as an expert and aggregates their outputs to exploit their respective knowledge, as illustrated in Figure 2(b). While MoLE alleviates information loss, it still faces two key challenges.

First, MoLE is coarse-grained, aggregating the entire output of each LoRA module without considering the heterogeneous contributions of individual rank-one components. To examine this limitation and the information loss induced by linear arithmetic, we evaluate LLaMA2-7B (Touvron et al., 2023) on two skill-composition tasks: Chinese mathematical reasoning (Chinese + Math) and Chinese code generation (Chinese + Code). For benchmarking, we propose a fine-grained approach, **Mixture of Rank-Wise Experts** (MoRE), which treats each rank-one component as an independent

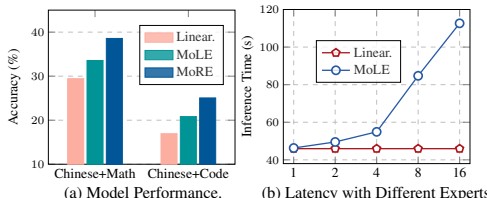

Figure 3: Performance and inference latency comparison of different knowledge reuse strategies.

expert (Figure 2(c); see Section 3.1 for details). Figure 3(a) shows that MoLE outperforms linear arithmetic by 4.13% and 3.88% on the two tasks, corroborating the hypothesis that parameter merging incurs information loss. However, MoLE still lags behind MoRE by 5.01% and 4.24%, underscoring its inability to fully exploit fine-grained knowledge embedded in rank-one components.

Second, MoLE incurs substantial computational overhead as the number of retrieved LoRA modules increases, severely limiting scalability in resource-constrained settings. We quantify this by measuring per-sample inference latency with varying numbers of integrated modules. Figure 3(b) shows that MoLE's latency rises sharply with more modules, whereas linear arithmetic maintains constant cost. For example, when integrating eight modules, MoLE incurs up to 1.84× higher latency than linear arithmetic. These findings highlight the performance and scalability bottlenecks of existing methods, motivating a more efficient, fine-grained knowledge reuse framework.

### 2.3 HETEROGENEOUS IMPORTANCE OF RANK-WISE EXPERTS

While MoRE enables fine-grained knowledge reuse, activating all experts still incurs substantial computational overhead. A practical way to meet resource budget is to sparsely activate rank-wise experts. However, optimally allocating expert quotas across matrices is non-trivial. Inspired by

(a) Importance across tasks.   (b) Importance across matrix types.   (c) Importance across layers.

Figure 4: Heterogeneous importance of rank-wise experts. (a) Importance distribution of the first-layer `Query` matrix for Chinese and Math LoRA modules. (b) Importance distribution of the first-layer `Query` and `Value` matrices for the Math LoRA module. (c) Importance distribution of the `Query` matrix across the first and second layers (Math LoRA).

AdaLoRA (Zhang et al., 2023b), which shows that parameter matrices contribute unevenly to fine-tuning performance, we hypothesize that rank-wise experts from different LoRA modules also vary in importance, and that allocating expert capacity accordingly can improve model performance.

To verify this, we evaluate LLaMA2-7B on the Chinese mathematical reasoning task, analyzing the importance distribution of rank-wise experts. We further investigate the effect of non-uniform expert allocation by varying quotas across different matrix types (e.g., `Query`, `Value`) with settings $\{16, 32, 48, 64\}$. For the rank-wise expert $E_m$, we quantify its importance by measuring the deviation between its output and the input representation. Given an input $\mathbf{h} \in \mathbb{R}^d$, the expert output is $\mathbf{h}_{E_m} = E_m(\mathbf{h})$, and the importance score is defined as:

$$s_{E_m} = 1 - \cos(\mathbf{h}, \mathbf{h}_{E_m}) = 1 - \frac{\mathbf{h}^\top \mathbf{h}_{E_m}}{\|\mathbf{h}\| \cdot \|\mathbf{h}_{E_m}\|}, \tag{6}$$

where $\cos(\cdot)$ denotes cosine similarity. Higher scores indicate that the expert contributes more to refining the representation. Figure 4(a) plots the importance distribution of 64 rank-wise experts from two task-specific LoRA modules in the first-layer `Query` matrix, revealing distinct importance levels both across tasks and within the same task. Figure 4(b) presents the importance distribution of 64 rank-wise experts from the Math LoRA module in the first-layer `Query` and `Value` matrices, indicating notable differences across matrix types. Figure 4(c) illustrates the importance distribution of 64 rank-wise experts from the Math LoRA module in the `Query` matrices of the first and second layers, highlighting substantial variation across layers.

Moreover, Figure 5 shows how model performance varies with the number of activated experts allocated to the `Query` and `Value` matrices. Model accuracy consistently improves as more experts are activated, with larger gains observed for the `Query` matrix than for the `Value` matrix, highlighting the differing sensitivities of different matrix types to expert capacity. Taken together, these results demonstrate that rank-wise experts exhibit heterogeneous importance across tasks, parameter matrix types, and layers, and that allocating expert quotas accordingly can enhance model performance without incurring additional computational overhead.

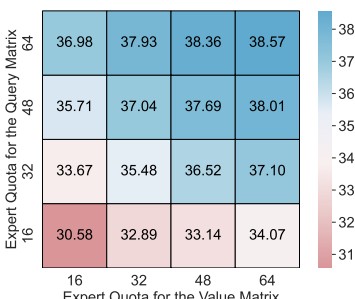

Figure 5: Impact of expert quota allocation on model performance.

## 3  OUR METHOD: SMARTFED

### 3.1  MIXTURE OF RANK-WISE EXPERTS

To enable effective reuse of existing LoRA knowledge while ensuring scalability and computational efficiency, we propose the Mixture of Rank-Wise Experts (MoRE). MoRE decomposes each LoRA module along the rank dimension into fine-grained, lightweight rank-wise components, and dynamically activates them through sparse, input-conditioned routing. Specifically, the low-rank update

Figure 6: Workflow of the Mixture of Rank-Wise Experts (MoRE). The server first retrieves relevant LoRA modules based on task requirements, then decomposes these modules into rank-wise components, and finally assembles these components into MoRE.

$\Delta \mathbf{W}\mathbf{x} = \mathbf{B}\mathbf{A}\mathbf{x}$ can be rewritten as:

$$\Delta \mathbf{W}\mathbf{x} = \sum_{i=1}^{r} \mathbf{B}_{:,i} \left( \mathbf{A}_{i,:}\mathbf{x} \right), \tag{7}$$

where $r$ is the LoRA rank, and $\mathbf{B}_{:,i} \in \mathbb{R}^{d \times 1}$ and $\mathbf{A}_{i,:} \in \mathbb{R}^{1 \times d}$ denote the $i$-th column of $\mathbf{B}$ and row of $\mathbf{A}$, respectively. Each term $\mathbf{B}_{:,i}\mathbf{A}_{i,:}$ constitutes a rank-one projection, which we treat as an independent **rank-wise expert**, formally defined as: $E_i(\mathbf{x}) = (\mathbf{B}_{:,i}\mathbf{A}_{i,:})\mathbf{x}$, where $i \in [1, r]$.

Unlike MoLE, MoRE decomposes LoRA modules into lightweight and interpretable experts, enabling fine-grained knowledge reuse and flexible cross-task information fusion. Figure 6 illustrates the construction workflow of MoRE. On the server side, task-relevant LoRA modules are first retrieved from a public repository based on the current task requirements. These modules are then decomposed into rank-wise experts, which are subsequently assembled to form the MoRE architecture. Given $N$ LoRA modules with ranks $\{r_1, \dots, r_N\}$, the expert pool contains $M = \sum_{n=1}^{N} r_n$ rank-wise experts. To enable input-aware knowledge fusion, we introduce a router that dynamically assigns weighting coefficients to experts conditioned on the input $\mathbf{x}$:

$$\mathbf{g} = \mathrm{softmax}(\mathbf{W}_{\mathrm{router}}\mathbf{x}), \tag{8}$$

where $\mathbf{W}_{\mathrm{router}} \in \mathbb{R}^{M \times d}$ is a learnable parameter matrix. To ensure computational efficiency, we sparsely activate only the top-$K$ experts for each token, with $K$ determined by hardware resources. This is implemented using a binary mask $\mathbf{m} \in \{0, 1\}^M$, where $\mathbf{m}_m = 1$ if $\mathbf{g}_m$ is among the top-$K$ entries in $\mathbf{g}$, and $\mathbf{m}_m = 0$ otherwise. The gated expert weights are then computed as:

$$\tilde{\mathbf{g}} = \mathbf{g} \odot \mathbf{m}, \tag{9}$$

where $\odot$ denotes the Hadamard product. The final output consists of the base transformation and the contributions from the sparsely activated rank-wise experts:

$$\mathbf{h}' = \mathbf{W}_0\mathbf{x} + \sum_{m=1}^{M} \tilde{\mathbf{g}}_m \cdot E_m(\mathbf{x}), \tag{10}$$

where $\tilde{\mathbf{g}}_m$ is the masked gating score for the $m$-th expert $E_m$. MoRE reframes federated fine-tuning from training LoRA from scratch into a modular, input-conditioned expert selection problem, substantially reducing the resource overhead of on-device training. Compared to prior knowledge reuse methods, MoRE offers two key advantages: 1) it decomposes fine-tuned LoRA modules into rank-wise experts, enabling fine-grained knowledge reuse across specific subspaces; and 2) it sparsely activates only the most relevant experts for each token, ensuring runtime efficiency and scalability.

## 3.2 ELASTIC EXPERT QUOTA ALLOCATION

Motivated by the observations in Section 2.3, we propose Elastic Expert Quota Allocation (EEQA), which adaptively determines the number of activated experts for each parameter matrix based on its contribution to model performance. The federated fine-tuning with EEQA proceeds as follows.

---

**Algorithm 1** The Elastic Expert Quota Allocation Strategy

---

**Require:** Normalized importance scores $\{\alpha_j\}_{j=1}^J$, total expert budget $B = K \cdot J$, and per-matrix upper bound $M$ on the number of activated experts.
**Ensure:** Expert quota for each matrix: $\{q_j\}_{j=1}^J$.

1: **Phase 1: Initial Allocation**
2: **for** $j = 1$ **to** $J$ **do**
3:    $\tilde{q}_j \leftarrow \min\left(\lfloor \alpha_j \cdot B \rfloor, M\right)$
4: **end for**
5: $R \leftarrow B - \sum_{j=1}^J \tilde{q}_j$                 { Remaining Quota }

6: **Phase 2: Residual Allocation**
7: $\mathcal{S} \leftarrow \{j \mid \tilde{q}_j < M\}$
8: Sort $\mathcal{S}$ by $\alpha_j$ in descending order
9: **for** $j \in \mathcal{S}$ **do**
10:    **if** $R = 0$ **then**
11:       **break**
12:    **end if**
13:    $\Delta \leftarrow \min(M - \tilde{q}_j, \ R)$
14:    $\tilde{q}_j \leftarrow \tilde{q}_j + \Delta; \ R \leftarrow R - \Delta; \ q_j \leftarrow \tilde{q}_j$
15: **end for**

16: **Return** $\{q_j\}_{j=1}^J$

---

1) The server first retrieves task-relevant LoRA modules and assembles them into MoRE (Section 3.1). It then distributes the global model to all participating devices. 2) Devices compute the importance score for each expert using Equation 6 and transmit the results to the server for aggregation. 3) The server randomly samples a subset of devices and transmits the router to them. Initially, each parameter matrix is assigned the same quota of activated experts, denoted by $K$.

4) The selected devices update the router using their local data. After completing local training, each device computes the cumulative importance score $S_j^{\text{Sum}}$ ($j \in [1, J]$) for each parameter matrix by summing the importance scores of its activated experts (Equation 11). The updated router and the set of scores $\{S_j^{\text{Sum}}\}_{j=1}^J$ are then transmitted back to the server.

$$S_j^{\text{Sum}} = \sum_{m=1}^M \tilde{\mathbf{g}}_m \cdot s_{E_m}. \tag{11}$$

5) The server aggregates these updates via FedAvg (McMahan et al., 2017), then applies a softmax over $\{S_j^{\text{Sum}}\}_{j=1}^J$ to obtain normalized importance scores (Equation 12). Afterwards, expert quotas for each matrix are reallocated following Algorithm 1. 6) The server then selects a new subset of devices for the next round, resuming the training process from step 4 until model convergence.

$$\alpha_j = \frac{\exp(S_j^{\text{Sum}})}{\sum_{j'=1}^J \exp(S_{j'}^{\text{Sum}})}, \quad j = 1, \ldots, J. \tag{12}$$

## 4 EXPERIMENTS

### 4.1 EXPERIMENTAL SETUP

Following prior works (Wang et al., 2024a; Prabhakar et al., 2024), we evaluate SMARTFED on three LLMs: LLaMA2-7B, LLaMA2-13B (Touvron et al., 2023), and Qwen2-7B (Bai et al., 2023). Our experiments cover three skill-composition tasks: Chinese mathematical reasoning (Chinese+Math), Chinese code generation (Chinese+Code), and hard math–word problems (Math+Code). For baselines without LoRA knowledge reuse, we train LoRA from scratch on Math23K (Wang et al., 2017), DoIT (Song et al., 2025), and MathCodeInstruct (Wang et al., 2023). Evaluation is performed on MGSM (Shi et al., 2022), DoIT, and GSM-Hard (Gao et al., 2023). Table 1 summarizes the tasks, datasets, and evaluation metrics. Appendix A provides the acquisition process of skill-specific LoRA modules, while Appendix B presents additional implementation details.

Table 1: Overview of skill-composition tasks, corresponding datasets, and evaluation metrics.

| Task | Skills Composed | Training Dataset | Testing Dataset | Evaluation Metric |
|---|---|---|---|---|
| Chinese Mathematical Reasoning | {Chinese, Math} | Math23K | MGSM | Accuracy |
| Chinese Code Generation | {Chinese, Code} | DoIT | DoIT | Pass@1 |
| Hard Math-Word Problems | {Math, Code} | MathCodeInstruct | GSM-Hard | Execution Accuracy |

Table 2: Performance comparison of different methods on various skill-composition tasks across LLMs. "Activated Rank" refers to the average number of ranks activated for each parameter matrix.

| LLM | Method | MGSM | DoIT | GSM-Hard | Average | Activated Rank |
|---|---|---|---|---|---|---|
| LLaMA2-7B | FedIT | 30.97 | 44.96 | 56.17 | 44.03 ($\downarrow$ 6.72) | 32 |
| | FwdLLM | 32.35 | 47.18 | 57.61 | 45.71 ($\downarrow$ 5.04) | 32 |
| | DoFIT | 31.28 | 46.09 | 56.89 | 44.75 ($\downarrow$ 6.00) | 32 |
| | FedAdapter | 32.47 | 48.70 | 58.05 | 46.41 ($\downarrow$ 4.34) | 32 |
| | AdaLoRA | 33.69 | 50.13 | 59.15 | 47.66 ($\downarrow$ 3.09) | 32 |
| | Linear Arithmetic | 29.43 | 43.81 | 54.90 | 42.71 ($\downarrow$ 8.04) | 32 |
| | LoRAHub | 33.41 | 49.25 | 58.79 | 47.15 ($\downarrow$ 3.60) | 32 |
| | MoLE | 33.56 | 49.54 | 58.69 | 47.26 ($\downarrow$ 3.49) | 64 |
| | LoRA-Flow | 33.75 | 49.95 | 59.06 | 47.59 ($\downarrow$ 3.16) | 64 |
| | SMARTFED | **35.48** | **52.59** | **64.19** | **50.75** | 32 |
| LLaMA2-13B | FedIT | 43.68 | 55.00 | 63.87 | 54.18 ($\downarrow$ 8.77) | 32 |
| | FwdLLM | 47.11 | 56.92 | 66.58 | 56.87 ($\downarrow$ 6.08) | 32 |
| | DoFIT | 45.85 | 56.37 | 64.91 | 55.71 ($\downarrow$ 7.24) | 32 |
| | FedAdapter | 48.29 | 58.57 | 66.79 | 57.88 ($\downarrow$ 5.07) | 32 |
| | AdaLoRA | 50.91 | 60.41 | 68.84 | 60.05 ($\downarrow$ 2.90) | 32 |
| | Linear Arithmetic | 42.24 | 52.79 | 63.19 | 52.74 ($\downarrow$ 10.21) | 32 |
| | LoRAHub | 50.43 | 60.06 | 68.42 | 59.64 ($\downarrow$ 3.31) | 32 |
| | MoLE | 50.20 | 60.17 | 68.36 | 59.58 ($\downarrow$ 3.37) | 64 |
| | LoRA-Flow | 51.12 | 60.32 | 68.70 | 60.05 ($\downarrow$ 2.90) | 64 |
| | SMARTFED | **53.18** | **63.01** | **72.65** | **62.95** | 32 |
| Qwen2-7B | FedIT | 30.59 | 44.17 | 55.04 | 43.27 ($\downarrow$ 7.08) | 32 |
| | FwdLLM | 31.82 | 46.41 | 56.52 | 44.92 ($\downarrow$ 5.43) | 32 |
| | DoFIT | 30.74 | 45.39 | 55.90 | 44.01 ($\downarrow$ 6.34) | 32 |
| | FedAdapter | 31.99 | 48.00 | 57.21 | 45.73 ($\downarrow$ 4.62) | 32 |
| | AdaLoRA | 33.05 | 49.45 | 58.01 | 46.84 ($\downarrow$ 3.51) | 32 |
| | Linear Arithmetic | 29.10 | 43.05 | 53.79 | 41.98 ($\downarrow$ 8.37) | 32 |
| | LoRAHub | 33.04 | 48.30 | 57.74 | 46.36 ($\downarrow$ 3.99) | 32 |
| | MoLE | 33.11 | 48.79 | 57.81 | 46.57 ($\downarrow$ 3.78) | 64 |
| | LoRA-Flow | 33.19 | 49.23 | 58.24 | 46.89 ($\downarrow$ 3.46) | 64 |
| | SMARTFED | **35.31** | **52.11** | **63.62** | **50.35** | 32 |

## 4.2 BASELINES

We compare SMARTFED against two categories of baselines. **1) Knowledge-Free Methods**: These approaches train LoRA from scratch without leveraging prior knowledge. Representative methods include FedIT (Zhang et al., 2024), an instruction tuning method; FwdLLM (Xu et al., 2023), a backpropagation-free optimization method; DoFIT (Xu et al., 2024), a domain-aware adaptation method; and FedAdapter (Cai et al., 2023), a training-efficiency method. We further adapt AdaLoRA (Zhang et al., 2023b), a parameter budget allocation method, to the federated scenario.

**2) Knowledge-Reuse Methods**: Since no federated fine-tuning methods reuse LoRA knowledge for LLM adaptation, we adapt several centralized approaches to the federated setting. These include Linear Arithmetic (Zhang et al., 2023a) and LoRAHub (Huang et al., 2023), which merge multiple LoRA modules into a single composite module; and MoLE (Wu et al., 2024) and LoRA-Flow (Wang et al., 2024a), which dynamically activate LoRA experts for task adaptation.

## 4.3 PERFORMANCE EVALUATION

Table 2 presents the performance of all methods on diverse skill-composition tasks across different LLMs. SMARTFED consistently delivers the best performance, significantly surpassing all baselines.

On LLaMA2-7B, SMARTFED outperforms knowledge-free methods with gains of up to 4.51%, 7.63%, and 8.02% across the three tasks, underscoring the benefit of reusing existing LoRA knowledge. However, naive combinations of LoRAs remain inadequate: Linear Arithmetic and LoRAHub

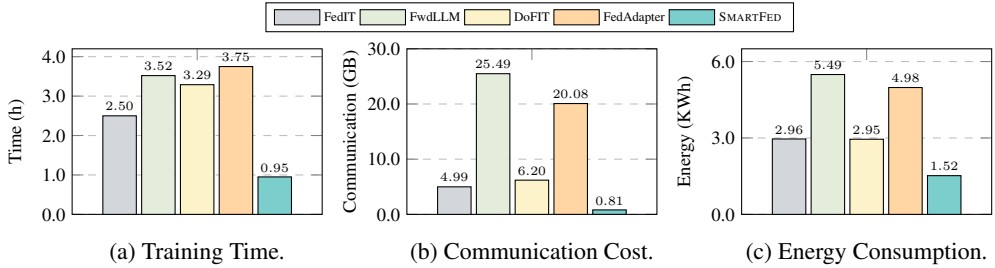

Figure 7: Overhead analysis of different methods on LLaMA2-7B for the Chinese+Math task.

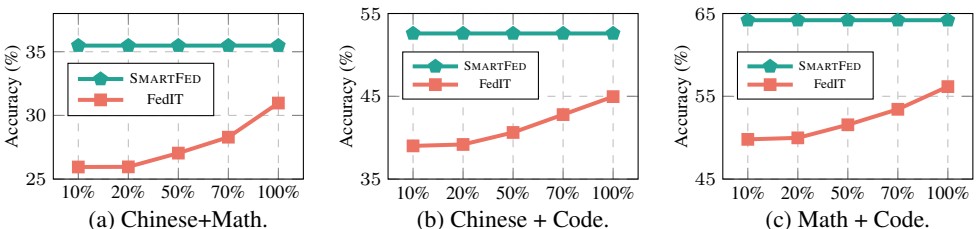

Figure 8: Performance comparison under varying fractions of training data on LLaMA2-7B. For example, "10%" denotes that each device uses only 10% of its local data for training.

suffer average performance drops of 8.04% and 3.60% relative to SMARTFED, respectively. These results suggest that direct parameter merging inevitably leads to information loss.

While MoLE and LoRA-Flow mitigate this issue through dynamic expert routing, their coarse-grained integration still incurs average performance drops of 3.49% and 3.16% relative to SMARTFED. Beyond this, SMARTFED exhibits remarkable generalization across model scales and architectures. On LLaMA2-13B, its advantage becomes even more pronounced, yielding average gains of up to 10.21%. Comparable improvements are also observed on Qwen2-7B, where SMARTFED achieves up to 8.37% average gains, underscoring its model-agnostic nature.

## 4.4 EFFICIENCY EVALUATION

We then evaluate the efficiency of SMARTFED along three dimensions: *training time*, *communication cost*, and *energy consumption*. All experiments are conducted on NVIDIA H800 GPUs, with energy consumption measured using `CodeCarbon` (Patterson et al., 2021). Figure 7 shows that SMARTFED substantially reduces resource overhead relative to baselines, accelerating model convergence by up to 3.95×, lowering communication cost by up to 31.47×, and reducing energy consumption by up to 3.61×. These improvements arise from SMARTFED 's reuse of task-adapted LoRA modules, which avoids costly retraining. Consequently, SMARTFED significantly enhances system efficiency, making federated fine-tuning practical in resource-constrained environments.

## 4.5 ADDITIONAL ANALYSIS

**Data Efficiency.** We further evaluate the data efficiency of SMARTFED on LLaMA2-7B under varying proportions of training data. As shown in Figure 8, SMARTFED consistently outperforms FedIT across all data regimes. Notably, with only 10% of the training data, SMARTFED surpasses FedIT trained on the full (100%) dataset by 4.51%, 7.63%, and 8.02% on the three tasks. These results highlight the remarkable data efficiency of SMARTFED, demonstrating its ability to achieve competitive or even superior performance under limited supervision—a property particularly valuable in data-scarce domains such as healthcare and biomedicine.

**Quota Allocation across Parameter Matrices.** To better understand the role of EEQA, we analyze the distribution of expert quotas across parameter matrices. As shown in Figure 9, the quota allocations exhibit significant variations across layers and matrix types. For example, the 16th-layer `Value` matrix is allocated 58 experts, whereas the corresponding `Query` matrices are assigned only one. This disparity demonstrates EEQA's ability to accurately identify and prioritize critical experts, thereby optimizing knowledge utilization without introducing additional computational overhead.

| | 1 | 2 | 3 | 4 | 5 | 6 | 7 | 8 | 9 | 10 | 11 | 12 | 13 | 14 | 15 | 16 | 17 | 18 | 19 | 20 | 21 | 22 | 23 | 24 | 25 | 26 | 27 | 28 | 29 | 30 | 31 | 32 |
|-------|---|---|---|---|---|---|---|---|---|----|----|----|----|----|----|----|----|----|----|----|----|----|----|----|----|----|----|----|----|----|----|----|
| Query | 52 | 44 | 46 | 61 | 4 | 40 | 23 | 53 | 31 | 52 | 15 | 50 | 2 | 62 | 3 | 1 | 32 | 56 | 63 | 51 | 33 | 18 | 14 | 3 | 28 | 9 | 34 | 24 | 60 | 17 | 4 | 39 |
| Value | 7 | 12 | 6 | 24 | 33 | 26 | 62 | 43 | 56 | 28 | 53 | 18 | 16 | 44 | 24 | 58 | 45 | 2 | 35 | 60 | 51 | 40 | 21 | 3 | 57 | 45 | 37 | 9 | 48 | 1 | 9 | 51 |

Layer Index

Figure 9: Distribution of expert quotas in LLaMA2-7B for the Chinese+Math task.

**Fine-Grained Knowledge Fusion.** To better understand MoRE's effectiveness, we visualize the average fusion weights of rank-wise components using a sample from the Chinese+Math task. As shown in Figure 10, MoLE assigns identical weights to all rank-wise components within each LoRA module (0.36 for Chinese LoRA and 0.64 for Math LoRA). In contrast, MoRE allocates weights independently to them, enabling more granular exploitation of task-relevant knowledge. This fine-grained control facilitates precise knowledge integration, which translates into superior performance.

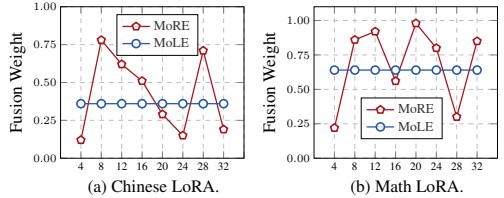

(a) Chinese LoRA.   (b) Math LoRA.

Figure 10: Average fusion weights for the `Query` matrix in the third layer of LLaMA2-7B.

### 4.6 ABLATION STUDY

Finally, we conduct extensive ablation studies to assess the effectiveness of the two key techniques in SMARTFED: MoRE and EEQA. Table 3 shows that both techniques make substantial contributions to model performance. On LLaMA2-7B, removing MoRE and EEQA leads to average performance drops of 4.23% and 3.45%, respectively, with similar degradation observed on LLaMA2-13B (4.95% and 3.31%). These results underscore that MoRE effectively leverages fine-grained knowledge from rank-wise experts, while EEQA adaptively prioritizes more influential experts. This synergy consistently enhances overall performance.

Table 3: Ablation study of SMARTFED.

| Method | Evaluation Task | | | Average |
|--------|------|------|----------|---------|
| | MGSM | DoIT | GSM-Hard | |
| **LLaMA2-7B** | | | | |
| SMARTFED | **35.48** | **52.59** | **64.19** | **50.75** |
| w/o MoRE | 32.52 | 48.81 | 58.22 | 46.52 (-4.23%) |
| w/o EEQA | 33.60 | 49.59 | 58.70 | 47.30 (-3.45%) |
| **LLaMA2-13B** | | | | |
| SMARTFED | **53.18** | **63.01** | **72.65** | **62.95** |
| w/o MoRE | 48.32 | 58.71 | 66.95 | 58.00 (-4.95%) |
| w/o EEQA | 50.25 | 60.20 | 68.46 | 59.64 (-3.31%) |

## 5 RELATED WORK

Recent studies have explored the integration of LoRA into federated fine-tuning pipelines. For instance, FedIT (Zhang et al., 2024) combines FedAvg with LoRA for instruction tuning. FLoRA (Wang et al., 2024b) and HETLoRA (Cho et al., 2024) address rank heterogeneity across devices through stacking-based and zero-padding strategies, respectively. However, these methods require training LoRA from scratch, imposing substantial overhead on edge devices.

Several approaches attempt to reduce computational costs by reusing knowledge from existing LoRA modules. LoRAHub (Huang et al., 2023) merges multiple LoRA modules into a composite module to achieve multi-task capabilities. MoLE (Wu et al., 2024) employs a router to integrate knowledge from different LoRA modules. However, they overlook data privacy concerns and fail to effectively utilize the knowledge. Departing from prior designs, SMARTFED decomposes LoRA into rank-wise experts for fine-grained knowledge reuse while preserving data privacy.

## 6 CONCLUSION

In this paper, we propose SMARTFED, a resource-efficient federated fine-tuning framework that reuses existing LoRA knowledge for LLM adaptation. Central to SMARTFED is the Mixture of Rank-Wise Experts, which decomposes LoRA modules into lightweight experts and activates them via a sparse, input-conditioned router. An Elastic Expert Quota Allocation strategy further improves knowledge utilization. Extensive experiments on multiple benchmarks demonstrate that SMARTFED consistently outperforms prior methods by a large margin.

## REPRODUCIBILITY STATEMENT

We place strong emphasis on the transparency and reproducibility of our work. To facilitate independent verification, Section 4.1 outlines all models, datasets, and evaluation metrics used in our experiments. For further clarity, Appendix A elaborates on the details of acquiring task-specific LoRA modules, while Appendix B documents the full set of hyperparameter choices and additional experimental details. Together, these resources ensure that our results can be reliably replicated and extended in future research.

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

# A  ACQUISITION OF SKILL-SPECIFIC LoRA MODULES

In this section, we detail the task-specific LoRA modules employed in our experiments. For different skill-composition tasks, we utilize different types of LoRA modules:

- Chinese+Math task: Chinese Chat LoRA and English Math LoRA;
- Chinese+Code task: Chinese Chat LoRA and English Code LoRA;
- Math+Code task: English Math LoRA and English Code LoRA.

The acquisition process for these LoRA modules is as follows:

- **Chinese Chat LoRA:** We utilize the dataset released by (Lai et al., 2023), comprising 52K training examples, to train a LoRA module capable of understanding and generating Chinese text.
- **English Math LoRA:** This LoRA module is trained on a dataset of 395K mathematical problems in English, constructed by (Yu et al., 2023).
- **English Code LoRA:** We train this LoRA module using the Magicoder dataset (Wei et al., 2023), which contains 186K code generation problems in English.

We integrate LoRA into the `Query` and `Value` matrices within attention modules. The LoRA rank $r$ is set to 32 and the scaling factor $\alpha$ is set to 64. We use the cosine warmup schedule and the peak learning rate is 1e-4. Each LoRA module is trained for 3 epochs with a warmup ratio of 0.04.

# B  MORE IMPLEMENTATION DETAILS

Following OpenFedLLM (Ye et al., 2024), we establish a client pool comprising 20 devices. For the Chinese mathematical reasoning task (Chinese+Math), we partition the Math23K dataset (Wang et al., 2017) across these devices, allocating approximately 1K samples to each device. For the Chinese code generation task (Chinese+Code), we partition the DoIT dataset (Song et al., 2025) across the devices. To ensure a fair comparison, we expand DoIT (Song et al., 2025) to 20K samples through web crawling, followed by manual verification using GPT-4 (Achiam et al., 2023), providing sufficient training data for participating devices. This expanded dataset is then evenly distributed, resulting in approximately 1K samples per device. Similarly, for experiments on hard math–word problems (Math+Code), we partition the MathCodeInstruct dataset (Wang et al., 2023) among the devices, with each device receiving approximately 1K samples.

In each training round, we randomly sample 10% of the devices to participate in the federated fine-tuning process, with each device performing 10 local update steps. The training process continues for 20 rounds. The batch size is set to 16, the learning rate is 5e-4, and $K$ is set to 32. For the Chinese mathematical reasoning task, we use MGSM (Shi et al., 2022) as the testing set. For the Chinese code generation task, we use DoIT (Song et al., 2025) as the testing set. For the hard math-word problems, we use GSM-Hard (Gao et al., 2023) as the testing set. We report the accuracy on MGSM, pass@1 on DoIT, and execution accuracy on GSM-Hard. For other baselines, the federated fine-tuning process continues until model convergence is achieved.

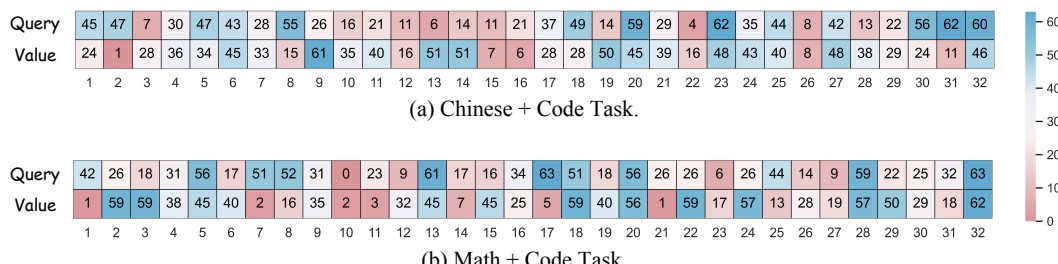

Figure 11: Distribution of expert quotas across different tasks in LLaMA2-7B.

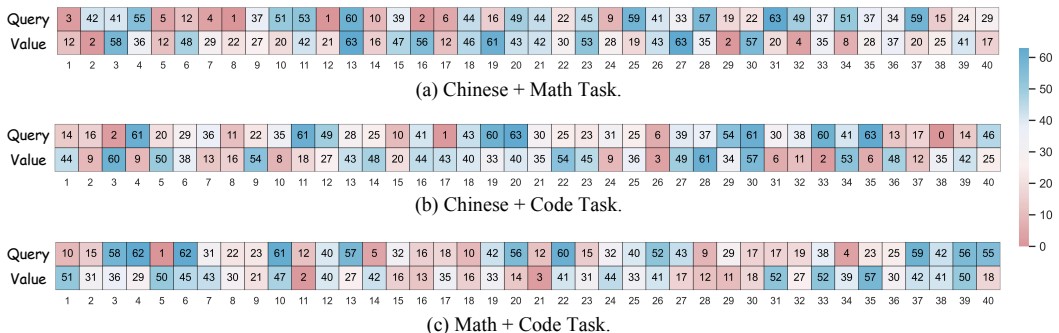

Figure 12: Distribution of expert quotas across different tasks in LLaMA2-13B.

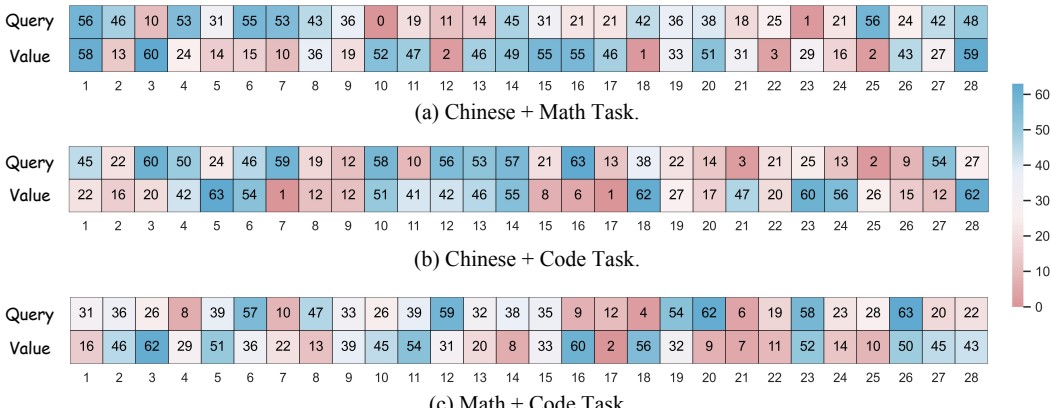

Figure 13: Distribution of expert quotas across different tasks in Qwen2-7B.

## C MORE EXPERIMENTAL RESULTS

### C.1 QUOTA ALLOCATION ACROSS PARAMETER MATRICES

In this section, we present comprehensive analyses of expert quota allocation patterns across different parameter matrices under various models and tasks. Figure 11 illustrates the expert quota distribution for LLaMA2-7B on Chinese+Code and Math+Code tasks, while Figure 12 and Figure 13 demonstrate the allocation patterns for LLaMA2-13B and Qwen2-7B across three tasks, respectively. Our analysis reveals distinct distribution patterns that vary significantly across parameter matrix types, layers, tasks, and model architectures. These heterogeneous allocation patterns validate the effectiveness of our proposed EEQA, which adaptively identifies crucial experts and consequently enhances knowledge utilization.

👤 User Input

> Carl's favorite food is cheese. This week, he eats a sandwich for lunch every day, using 2 slices of cheese per sandwich. On 3 days of the week, he has cheese and omelet for breakfast, with each omelet using 1 more slice of cheese than each sandwich. He made a large portion of mac and cheese for several dinners this week, using 8 slices of cheese. **How many slices of cheese did he use in total?**

- - - - - - - - - - - - - - - - - - - - - - - - - - - - - - - - - - - - - - - - - -

SmartFed Output:

> Carl eats sandwiches all 7 days of the week, using 2 slices of cheese per sandwich, so sandwiches used 7 * 2 = 14 slices of cheese in total. He had omelets on 3 days, with each omelet using 1 more slice than a sandwich, which means 2 + 1 = 3 slices, so omelets used 3 * 3 = 9 slices of cheese in total. The mac and cheese used 8 slices of cheese. In total, he used 14 + 9 + 8 = 31 slices of cheese this week. \n#### 31\n The answer is: 31. 👍✨

MoLE Output:

> Carl eats sandwiches 5 days a week, using 2 slices of cheese per sandwich, so sandwiches used 5 * 2 = 10 slices of cheese in total. He had omelets on 3 days, with each omelet using 1 more slice than a sandwich, which means 2 + 1 = 3 slices, so omelets used 3 * 3 = 9 slices of cheese in total. The mac and cheese used 8 slices of cheese. In total, he used 10 + 9 + 8 = 27 slices of cheese this week. \n#### 27\n The answer is: 27 ❌

Figure 14: Case study on the Chinese+Math task (LLaMA2-7B). Chinese query and responses are translated to English for presentation clarity.

## C.2 CASE STUDY

To further demonstrate the effectiveness of SMARTFED, we present a case study on the Chinese+Math task using LLaMA2-7B (Figure 14). This example illustrates how fine-grained knowledge fusion enables more precise comprehension of user queries. Unlike MoLE, which treats each LoRA module as a whole and assigns uniform weights, SMARTFED employs rank-wise experts to selectively integrate knowledge from task-related subspaces. This enhanced granularity allows SMARTFED to capture subtle semantic nuances, resulting in precise solutions. In contrast, MoLE exhibits significant hallucinations—notably misinterpreting fundamental facts like the number of days in a week—highlighting the limitations of coarse-grained fusion approaches.

Furthermore, Figure 16 presents an illustrative example from the Math+Code task, where Figure 15 demonstrates the average fusion weights of rank-wise components for this sample. We observe that MoLE fails to retain crucial user-provided information, resulting in inaccurate auxiliary program generation and consequently erroneous solutions. In contrast, SMARTFED, through a more flexible knowledge reuse strategy, exhibits enhanced comprehension of user requirements and robust information retention.

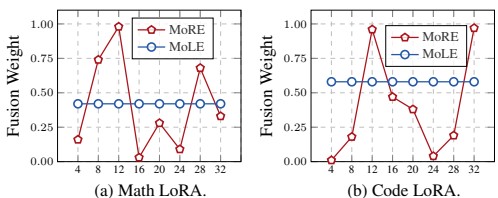

(a) Math LoRA.      (b) Code LoRA.

Figure 15: Average fusion weights of rank-wise components in the `Query` matrix at the third layer for the Math+Code task (LLaMA2-7B).

This, in turn, facilitates the generation of precise programs and accurate solutions, demonstrating SMARTFED's capability to handle complex mathematical problems with high fidelity. Overall, these results underscore the advantages of SMARTFED in delivering more reliable and precise responses through fine-grained knowledge integration.

## D THE USE OF LARGE LANGUAGE MODELS

During manuscript preparation, a large language model (LLM) was occasionally employed as an auxiliary assistant to refine language expression, such as improving sentence fluency and enhancing readability. The model was not involved in generating original research contributions: it did not participate in formulating research questions, designing methodologies, conducting experiments, analyzing results, or drafting substantive scientific content. All core intellectual work, including the

🧑 User Input

Martha is knitting winter wear for her 2189997 grandchildren. They're triplets, so they're all the same size. She wants to make a hat, scarf, sweater, mittens, and socks for each of them. It takes 2 skeins of wool to make a hat, 4 for a scarf, 12 for a sweater, 1 for a pair of mittens, and 2 for a pair of socks. **How many skeins of wool will she need to buy?**

**SmartFed Output:**

```
Code:"
def solution():
    \"\"\"Martha is knitting winter wear for her 2189997
grandchildren. They're triplets, so they're all the same
size. She wants to make a hat, scarf, sweater, mittens,
and socks for each of them. It takes 2 skeins of wool to
make a hat, 4 for a scarf, 12 for a sweater, 1 for a pair
of mittens, and 2 for a pair of socks. How many skeins
of wool will she need to buy?\"\"\"
    num_grandchildren = 2189997
    skeins_per_hat = 2
    skeins_per_scarf = 4
    skeins_per_sweater = 12
    skeins_per_mittens = 1
    skeins_per_socks = 2
    skeins_per_grandchild = skeins_per_hat +
skeins_per_scarf + skeins_per_sweater +
skeins_per_mittens + skeins_per_socks

    total_skeins = num_grandchildren *
skeins_per_grandchild

    result = total_skeins
    return result
"
```
**Answer: 45989937.0** 👍

**MoLE Output:**

```
Code:"
def solution():
    \"\"\"Martha is knitting winter wear for her 2189997
grandchildren. They're triplets, so they're all the same
size. She wants to make a hat, scarf, sweater, mittens,
and socks for each of them. It takes 2 skeins of wool to
make a hat, 4 for a scarf, 12 for a sweater, 1 for a pair
of mittens, and 2 for a pair of socks. How many skeins
of wool will she need to buy?\"\"\"
    num_grandchildren = 2189997
    skeins_per_hat = 2
    skeins_per_scarf = 4
    skeins_per_sweater = 12
    skeins_per_mittens = 1

skeins_per_grandchild = skeins_per_hat +
skeins_per_scarf + skeins_per_sweater +
skeins_per_mittens

    total_skeins = num_grandchildren *
skeins_per_grandchild

    result = total_skeins
    return result
"
```
**Answer:41609943.0** ❌

Figure 16: Case study on the Math+Code task (LLaMA2-7B).

development of ideas, execution of experiments, and interpretation of findings, was carried out independently by the authors. Any linguistic suggestions offered by the LLM were critically reviewed and selectively incorporated, ensuring that accuracy, originality, and scholarly integrity were fully maintained. The authors alone bear responsibility for the research content and conclusions, and the LLM is not listed as a contributor or author.

