# OpenReview forum: "Elastic Mixture of Rank-Wise Experts for Knowledge Reuse in Federated Fine-Tuning"
_ICLR.cc/2026/Conference — ICLR 2026 Conference Withdrawn Submission_

### Official Review · Reviewer_QKU2 · 2025-10-30

**Soundness:** 3
**Presentation:** 4
**Contribution:** 2
**Rating:** 4
**Confidence:** 4

**Summary:**

The paper proposes SMARTFED, a resource-efficient framework for federated fine-tuning that reuses existing LoRA modules instead of training LoRA from scratch. The core ideas are: (1) a Mixture of Rank-Wise Experts that decomposes each LoRA into rank-1 “experts” and routes them sparsely per token for fine-grained knowledge reuse; and (2) Elastic Expert Quota Allocation, which adaptively allocates the number of active rank-wise experts per parameter matrix based on learned importance. Across LLaMA2-7B/13B and Qwen2-7B on three “skill-composition” tasks, SMARTFED reports higher accuracy, faster convergence, and lower communication/energy cost than baselines.

**Strengths:**

1. Rank-wise expertization of LoRA for reuse (not retraining) is a crisp idea that avoids cross-task interference from naive merging and improves upon coarse MoLE-style routing.
2. Careful ablations show both MoRE and EEQA matter. Efficiency analyses quantify wall-clock, communication, and energy benefits.
3. Demonstrated improvements across models and tasks, including data-efficiency under 10% data regimes, are compelling for realistic federated scenarios.

**Weaknesses:**

1. The method presumes relevant, high-quality task LoRAs can be found and that their ranks or placements suit the target task.
2. Practical latency on device for per-token top-K over large expert pools (sum of ranks across many LoRAs) is not deeply profiled.
3. Experiments focus on LLaMA2-7B/13B and Qwen2-7B, which are now relatively dated choices for open LLM backbones.

**Questions:**

1. How sensitive is performance to retrieval errors (irrelevant or partially relevant LoRAs)?
2. How frequently do you recompute the importance score in practice under non-IID drift? What is the communication overhead for transmitting importance stats per round?

---

### Official Review · Reviewer_Ubny · 2025-10-30

**Soundness:** 3
**Presentation:** 3
**Contribution:** 2
**Rating:** 6
**Confidence:** 5

**Summary:**

This paper introduces SMARTFED, a federated fine-tuning approach for LLMs that reuses existing LoRA modules and trains only a lightweight router to select and combine them based on input semantics. Its MoRE design decomposes each LoRA into rank-1 experts and activates only the top-K per token, while EEQA adaptively allocates expert capacity to more important matrices/layers. Across skill-composition tasks (language × math/code) and multiple base models, SMARTFED achieves higher accuracy than training new LoRAs or coarse expert methods, while greatly reducing training time, communication, and energy—and it can outperform strong baselines even with just 10% of the data.

**Strengths:**

Creative reuse of LoRA: the router-based composition cleverly exploits LoRA’s rank-wise decomposability to mix skills without retraining full adapters.

No from-scratch LoRA training: this slashes compute and communication for federated learning, making deployment far more practical on edge devices.

Modular and scalable: sparse rank-wise activation plus adaptive quotas yields strong data efficiency and plug-and-play growth with LoRA libraries, improving latency and energy in real systems.

**Weaknesses:**

Heavy reliance on the quality and coverage of existing LoRA modules. If client data are private or domain specific and no matching public LoRA exists, the method may misalign with the goals of federated learning; the experiments do not cover fully private, no-LoRA settings.

System and tuning complexity. The router and EEQA require importance scoring, quota allocation, and Top K choices; performance may be sensitive to hyperparameters and client heterogeneity, and the paper offers limited robustness and failure-mode ablations.

Repository and security scalability. Real deployments must curate, distribute, and govern large LoRA libraries, which increases storage and latency costs, raises licensing and provenance questions, and exposes the system to poisoned or low-quality adapters that can bias routing.

**Questions:**

Please see the weakness

---

### Official Review · Reviewer_jfGt · 2025-11-04

**Soundness:** 2
**Presentation:** 2
**Contribution:** 2
**Rating:** 2
**Confidence:** 5

**Summary:**

This paper proposes SMARTFED, a resource-efficient federated fine-tuning framework that reuses existing LoRA modules through a novel Mixture of Rank-Wise Experts (MoRE) mechanism and adaptive Elastic Expert Quota Allocation (EEQA) strategy.

**Strengths:**

This is an interesting paradigm, but it requires more comprehensive evaluation to fully validate its generality and robustness.

**Weaknesses:**

1. Figure 4 lacks clarity and insight. The observations in Figure 4 are not intuitive and provide limited insight, as they are only evaluated on one or two layers.

2. On the optimal number of experts. Given the redundancy within LLMs, is there an optimal number of experts? I am skeptical whether the results shown in Figure 5 would generalize across different tasks.

3. Router-only training raises concerns on generalization. In the proposed method, only the router is trained. It is unclear whether this approach can effectively handle long-tail or out-of-distribution (OOD) data. Specifically, how can open-sourced LoRA modules adapt to such data by merely tuning the router, especially when no new knowledge is incorporated into the adapters themselves? Please elaborate on how retrieval works in an open-world setting and clarify whether the experimental setup accounts for OOD scenarios.

4. Lack of explanation on the decomposition strategy. The decomposition process is not clearly explained. Is the splitting simply done empirically by columns? More details are needed.

5. Structural issues in organization and flow. The overall structure of the paper needs improvement. For example, Section 2.2 discusses results on MoRE before introducing the method itself. Moreover, the importance score is introduced prematurely in Section 2.2 for preliminary evaluation and then used again in EEQA in Section 3.2, which may confuse readers.

6. Fairness of comparison and practical limitations. The fairness of the comparisons is questionable. If pre-trained LoRA modules are not publicly available, the proposed method may incur higher data acquisition and training costs, which could undermine its practical benefits and lead to greater resource consumption compared to existing approaches.

7. Can this method extend to tens or hundreds of tasks?

**Questions:**

See Weaknesses.

---

### Official Review · Reviewer_NH6X · 2025-11-08

**Soundness:** 2
**Presentation:** 3
**Contribution:** 2
**Rating:** 4
**Confidence:** 4

**Summary:**

This paper proposes a framework named SmartFed, which aims to improve the efficiency of federated LLM fine-tuning at the edge. SmartFed consists of two key techniques, namely the mixture of rank-wise experts (MoRE) and Expert Quote Allocation (EQA). MoRE treats single ranks in LoRA weights as adapters, while EQA assigns ranks for different tasks based on their corresponding importance. Experimental results demonstrate the performance of SmartFed, which appears to outperform existing benchmarks by a noticeable margin.

**Strengths:**

- The paper is well written and well motivated.
- Improving the efficiency of federated learning at the edge appears to be a promising direction to explore.
- The proposed SmartFed method is presented clearly; it is intuitive and easy to follow.
- The experimental results seem reasonable.

**Weaknesses:**

- Using PEFT for federated learning is a somewhat crowded research area. Many methods have already been developed, as shown in the paper (with numerous baseline methods compared). Looking at the main results in Table 2, SmartFed appears to outperform other noticeable baselines, but it is difficult to assess the statistical significance of these improvements. For instance, how meaningful is an improvement of about two points on LLaMA2 over some benchmarks? That being said, SmartFed seems to make only a marginal contribution to the field.

- Only outdated LLMs have been studied. We already have LLaMA 4 and Qwen 3. Why are the experiments still conducted on LLaMA2 and Qwen2?

- The selected tasks do not seem very meaningful. Under which application scenarios would one want to fine-tune a model for better Chinese code generation using a federated approach?

- Some baselines are still missing (e.g., Flora, which is cited but not compared), even though many baselines have already been included. This further reinforces the impression that the research area is already quite crowded.

**Questions:**

- How does SmartFed perform on models such as Qwen 3, LLaMA 4, and GPT-OSS?
- What are the real application scenarios for federated fine-tuning at the edge? Frankly, I find it difficult to think of reasonable examples—perhaps some smart health scenarios?

---

### Note · Authors · 2025-12-31

I have read and agree with the venue's withdrawal policy on behalf of myself and my co-authors.